# Performance and Application of Commercially Available Loop-Mediated Isothermal Amplification (LAMP) Kits in Malaria Endemic and Non-Endemic Settings

**DOI:** 10.3390/diagnostics11020336

**Published:** 2021-02-18

**Authors:** Ulrika Morris, Berit Aydin-Schmidt

**Affiliations:** Department of Microbiology, Tumor, and Cell Biology, Karolinska Institutet, 17177 Stockholm, Sweden; berit.schmidt@ki.se

**Keywords:** loop-mediated isothermal amplification, LAMP, *Plasmodium*, malaria, low transmission, pre-elimination, malaria in pregnancy, non-endemic setting, returning travelers

## Abstract

Loop-mediated isothermal amplification (LAMP) is a sensitive molecular tool suitable for use as a near point-of-care test for the diagnosis of malaria. Recent meta-analyses have detailed high sensitivity and specificity of malaria LAMP when compared to microscopy, rapid diagnostic tests, and polymerase chain reaction in both endemic and non-endemic settings. Despite this, the use of malaria LAMP has primarily been limited to research settings to date. In this review, we aim to assess to what extent commercially available malaria LAMP kits have been applied in different settings, and to identify possible obstacles that may have hindered their use from being adopted further. In order to address this, we conducted a literature search in PubMed.gov using the search terms (((LAMP) OR (Loop-mediated isothermal amplification)) AND ((Malaria) OR (Plasmodium))). Focusing primarily on studies employing one of the commercially available kits, we then selected three key areas of LAMP application for further review: the performance and application of LAMP in malaria endemic settings including low transmission areas; LAMP for malaria screening during pregnancy; and malaria LAMP in returning travelers in non-endemic settings.

## 1. Introduction

Loop-mediated isothermal amplification (LAMP) is a sensitive molecular tool suitable for use as a near point-of-care test for the diagnosis of malaria in low resource settings [1]. The LAMP methodology—first published in year 2000—relies on isothermal deoxyribonucleic acid (DNA) amplification employing the *Bacillus stearothermophilus* (*Bst*) DNA polymerase with strand displacement activity [2]. LAMP can therefore be performed at a single temperature with a simple heating block or water bath, reducing the need for sensitive and expensive machinery such as polymerase chain reaction (PCR) thermocyclers. The LAMP reaction is primed by a specific set of four to six primers that identify distinct regions on the target DNA. The design of the primers results in DNA loop formations and several inverted repeats of the target DNA [2]. This autocycling strand-displacement DNA synthesis makes the amplification highly efficient and specific, allowing the amplification of a few DNA copies to 10^9^ copies under isothermal conditions in less than one hour, reducing the time-to-result [3,4]. DNA amplification can be detected by eye by a change in turbidity caused by white precipitate of magnesium pyrophosphate formed during the reaction, or under ultraviolet (UV) fluorescence if a fluorescent indicator such as calcein is added to the reagents [3]. Visual detection avoids the need for opening the reaction tube post-amplification, hence the reaction is conducted in a closed system which reduces the risk of DNA contamination [5]. Furthermore, the *Bst* polymerase is more robust towards inhibition than *Taq* polymerase in conventional PCR, making it suitable for use with simple and field friendly DNA extraction methods, but maintaining a sensitivity comparable to PCR [1,6,7,8,9].

The first malaria specific LAMP assay targeting the *Plasmodium falciparum* 18S ribosomal ribonucleic acid (rRNA) genes was published in 2006 [10]. This was followed by *Plasmodium* genus and species-specific LAMP assays published in 2007 [7]. Since then, over 40 different LAMP methods have been developed. These developments—some of which were recently reviewed [9,11]—have aimed at improving the sensitivity of the assay by targeting mitochondrial DNA [6] or alternative gene targets [12,13,14,15]. Other methods have aimed at improving or mitigating DNA extraction processes [16,17,18,19], or improving the read-out of the results by incorporating different dyes [20,21,22] or by combining LAMP with lateral flow dipsticks providing a similar result format to malaria rapid diagnostic tests (RDTs) [23].

Recent meta-analyses evaluating the diagnostic accuracy of LAMP for malaria have detailed high sensitivity and specificity of LAMP when compared to microscopy, PCR, and RDTs in both endemic and non-endemic settings (Table 1). The pooled sensitivity and specificity of LAMP has largely remained greater than 95% whichever the comparator [1,24,25], with an area under the curve of greater than 0.98 demonstrating that malaria LAMP is a test with excellent diagnostic performance [24,25]. These meta-analyses concluded that the LAMP method is a robust tool for diagnosing malaria in both symptomatic and asymptomatic individuals [24], and that LAMP is one of the most promising new diagnostic tools for use in malaria endemic settings [25].

In this review, we aim to assess to what extent commercially available malaria LAMP kits have been applied in different settings and to identify possible obstacles that may have hindered these kits from being adopted further. In order to address this, we conducted a literature search in PubMed.gov using the search terms (((LAMP) OR (Loop-mediated isothermal amplification)) AND ((Malaria) OR (*Plasmodium*))). The 148 publications identified to be directly related to *Plasmodium* detection by LAMP in humans were grouped according to commonly occurring key words and/or relevance (Appendix A). Focusing primarily on studies that have utilized one of the commercially available LAMP kits, we then selected three recently highlighted key areas of LAMP application [1,9] for further review: the performance and application of LAMP in malaria endemic settings including low-transmission areas; LAMP for malaria screening during pregnancy; and malaria LAMP in returning travelers in non-endemic settings.

## 2. Commercially Available LAMP Kits

Fifty-eight of the 148 identified malaria LAMP publications employed one of two brands of field-stable and CE-marked (where CE stands for Conformité Européenne” in French or “European Conformity” in English) malaria LAMP kits that are now commercially available: the Illumigene^®^ (now Alethia ^®^) malaria LAMP (Meridian Bioscience Inc., Cincinnati, OH, USA) and the Loopamp^TM^ Malaria Detection Kits (Eiken Chemical Co., Tokyo, Japan). Neither of the kits require a cold chain, as they contain ready to use lyophilized reagents that are stable at room temperature [9]. Both brands target the mitochondrial DNA of *Plasmodium* species for genus-level identification. In addition, the Loopamp Malaria Pf Detection Kit and the more recent Loopamp Malaria Pv Detection Kit can differentiate *P. falciparum* and *P. vivax* infections from other species, respectively (Table 2) [26].

Both the Illumigene and the Loopamp kits can be performed on fresh or frozen blood, in combination with simple and quick DNA extraction methods. The Loopamp kits can also be performed on dried blood spots [27,28,29]. Both brands provide high analytical sensitivity, with a time-to-result of 1–2 h [27,29,30,31]. Both kits are easy to handle but require some basic laboratory skills as they require blood sample preparation and measures to avoid DNA cross-contamination [32]. The risk for DNA contamination is however minimized, as both kits utilize a robust closed system, where tubes with amplified products are never opened (unlike conventional or nested PCR) [33]. A drawback of the LAMP methodology is that it does not provide parasite quantification.

### 2.1. The Illumigene^®^ (Alethia^®^) Malaria LAMP

The Illumigene Malaria LAMP kit is easy to perform, with DNA extraction and result readout included in the same kit. DNA extraction is conducted by a simple gravitation system that employs lysis buffers and columns to simplify sample preparation. In this approach, whole blood is mixed with a lysis buffer and either used directly in the LAMP assay (simple filtration assay) or passed through a column that purifies the DNA via gravity (Illumigene malaria PLUS) [31]. Both procedures rely on chemical lysis and produce amplifiable DNA within 10 minutes, requiring only the use of micropipettes, DNase/Rnase free sterile pipette tips, and latex gloves. Each test device consists of two tubes, a test tube with primers targeting the *Plasmodium* genus and a control tube with primers to detect a housekeeping human gene used as a DNA extraction and amplification control.

Result readout with an Illumigene Malaria Illumipro-10™ incubator/reader measures turbidity induced by the formation of magnesium pyrophosphate. Ten samples can be analyzed per run in this platform [34]. The detection limit of the assay is 2.0 parasites per microliter (p/µL) or 0.3 p/µL for *P. falciparum* (depending on which extraction method is used) and 0.1 p/µL for *P. vivax* [30,31]. A disadvantage with the kit is that no species differentiation is possible, and there are a few reports of occurrence of invalid results [32,34].

The Illumigene kit is preferably conducted on venous EDTA whole blood samples, making it possible to perform the LAMP assay with some delay. This blood sampling method can however be challenging in resource limiting settings [31]. Furthermore, the Illumigene kit has a high cost (~28 euros per sample plus the cost of the Illumipro-10™ incubator/reader), limiting its use to developed countries for diagnosis of imported malaria [9,30].

### 2.2. The Loopamp^TM^ Malaria Detection Kits

The Loopamp Malaria Detection Kits have been endorsed by the Foundation for Innovative New Diagnostics (FIND) who has developed standardized procedures for their use [11,35]. The Loopamp Malaria Pan Detection Kit is typically used for malaria screening with an analytical sensitivity of 1–2 p/µL [27,29,36,37]. Positive samples can thereafter be retested using the Pf or Pv detection kits for further species identification [38].

DNA can be extracted with the commercially available Loopamp PURE DNA Extraction Kit, consisting of a series of interlocking plastic components providing a closed system for preparation of a blood aliquot requiring only a heat block or incubator, micropipettes, DNase/Rnase free sterile pipette tips, and latex gloves [39]. The Loopamp kits can also be used with a cheaper, quick and easy, boil and spin DNA extraction method, but requires additional equipment such as a centrifuge [9,35]. The LAMP reaction can either be conducted in a simple heat block or water bath, or in a real-time turbidimeter. The formation of magnesium pyrophosphate resulting in turbidity, or fluorescence produced by the release of calcein upon amplification in positive samples, can immediately be read by eye or by using a simple UV light source, respectively [40]. However, the use of a UV-lamp for reading results is comparably less objective than a turbidimeter [36,41].

The LA-500 real-time turbidimeter (Eiken Chemical Co.) or the HumaLoop M (Human diagnostics Worldwide, Wiesbaden, Germany) platform for sample preparation, amplification, and easy visual result reading under a built in UV light can hold 16 tests per run. If a simple heating block system is used then batches of up to 46 samples can be run together with a positive and negative control (the maximum number of tubes that fit on a regular heat-block), making it a useful tool for prevalence surveys in endemic areas [38,42]. Furthermore, a high throughput DNA extraction platform that can extract up to 94 samples at a time has been assessed, albeit with varying performance [41,43]. The Loopamp kits cost about 5.20 euros per test, excluding the costs for DNA extraction and equipment.

## 3. Performance and Application of Commercial LAMP Kits in Malaria Endemic Settings

The performance of LAMP has been evaluated in malaria endemic field settings in over 30 publications, of which 15 studies employed the Loopamp^TM^ MALARIA Detection kits and two studies the Illumigene^®^ (Alethia) malaria LAMP. Overall, LAMP has shown ≥95% pooled sensitivity and specificity for detecting both *P. falciparum* and *P. vivax* infections when used in endemic settings [25]. The consensus among studies that have employed one of the commercial kits in malaria endemic settings is that the methods are easy to perform after only 3–5 days laboratory training [31,38,39,42,44,45,46,47,48,49], even in the remotest of field settings [47].

### 3.1. Performance of LAMP in Asymptomatic and Low-Density Infections

As malaria transmission declines in areas of successful malaria control, the relative proportion of asymptomatic and low-density malaria infections increases [50]. These infections are often subpatent, i.e., falling below the detection limit of conventional malaria diagnostic tools such as microscopy or RDTs. Hence, as malaria transmission decreases, new and more sensitive field applicable screening tools are needed for the detection and management of very low-density infections, especially if malaria elimination is to be considered possible [25,51].

The improved analytical sensitivity of LAMP, resulting in the detection of significantly more infections compared to RDT or microscopy [42,46,47,48,52,53], and the reduced time-to-result compared to PCR [31,38,39,41,42,47,48], has put LAMP forward as a promising tool for near point-of-care detection of low parasite density infections in asymptomatic carriers, especially in low endemic and pre-elimination areas [3,9,24,25]. However, despite its improved analytical sensitivity, LAMP may still miss a substantial proportion of the asymptomatic reservoir of very low-density infections [41,46,52], especially in low transmission settings (Table 3). Although LAMP may provide a better understanding of the prevalence and distribution of low-density asymptomatic infections than conventional diagnostic tools, a rapid turn-around time may not necessarily be the highest priority in malaria epidemiological surveys where highly sensitive PCR-based methods are still often the method of choice [54]. On the other hand, results should be made available within 48 hours of testing in detect-and-treat approaches such as reactive case detection (i.e., screening and treatment of household members and neighbors of passively identified index cases), and mass or focal testing and treatment strategies. Further studies are however needed to assess if employing LAMP will significantly improve the impact such resource-intensive elimination strategies have on malaria transmission [52,55].

### 3.2. Application of LAMP in Prevalence Surveys in Malaria Endemic Settings

Eight studies—all of which were conducted in sub-Saharan Africa—have applied commercial LAMP kits as a screening tool in malaria prevalence surveys. For instance, LAMP has been used in several studies in Uganda to investigate the effect of HIV infection on malaria incidence [56]; the effect of indoor residual spraying on the prevalence of asymptomatic infections [57]; for assessing the prevalence of microscopic and submicroscopic malaria infections in different transmission sites [58]; and the prevalence of submicroscopic infections among schoolchildren [59]. In these studies, dry blood spots on filter paper were collected at point-of-care. DNA extraction was conducted by a Chelex-based method, followed by the Loopamp Malaria Pan Detection Kit. Depending on age, prevalence of asymptomatic infections, and malaria transmission intensity, the LAMP positivity rate was two to ten times greater than the positivity rate by microscopy or RDT [56,57,58,59].

LAMP has also been applied for assessing risk factors and spatial clustering of asymptomatic malaria though active and reactive surveillance in low transmission settings [60,61,62]. In these studies, dry blood spots were collected for later Chelex extraction followed by screening with the Loopamp Pan Detection Kit. LAMP detected significantly more infections than RDTs, with clustering of asymptomatic low-density infections around index cases [60,62]. In the study by Smith et al. RDT identified only 17% of the index and neighbor cases detected by LAMP, suggesting that infections missed by RDT during reactive case detection may be responsible for 50–71% of transmission from humans to mosquitoes [62].

A single study performed in Congo has applied the Illumigene Malaria kit. Venous blood (4 mL) with EDTA was taken from 1088 children aged between 6 and 59 months for assessing the prevalence of anemia and its relationship with asymptomatic submicroscopic *Plasmodium* infection. Malaria prevalence was 16% and 34% by microscopy and LAMP, respectively and submicroscopic *Plasmodium* infection was found in 22% of the children [63].

## 4. Pros and Cons of Commercially Available LAMP Kits in Low Resource Settings

It is unlikely that LAMP will replace conventional diagnostic tools such as RDTs and microscopy in the point-of-care diagnosis of clinically symptomatic malaria in endemic settings [1,9,33,64]. LAMP in its current formats does not completely fulfil the World Health Organization (WHO) ASSURED criteria (i.e., being Affordable, Sensitive, Specific, User-friendly, Robust and rapid, Equipment free, and Deliverable) for identifying appropriate diagnostic tests for resource-constraint settings, and some improvements are likely to be necessary for this tool to become readily available [1,9]. Whilst the commercially available diagnostic kits certainly offer user-friendliness, with high specificity and robust, rapid, and deliverable assays that are independent of a cold chain, they are not equipment free and require a reliable source of electricity. Despite the ease of use, LAMP still requires good enough facilities for preparing basic molecular assays, ideally with separate workstations for avoiding DNA-cross-contamination [64]. In addition, the technicians running the assays require sufficient knowledge to manage DNA contamination, an issue with all nucleic acid amplification-based methods that needs to be addressed [28,36,42,45,65]. Efforts to simplify DNA isolation methods [19], or reduce the requirement of electricity dependent equipment [45], have succeeded only on behalf of the sensitivity or specificity of the assay. In addition, the high cost of these commercially available assays has been put forward as a limiting factor in their uptake [33,44].

### 4.1. Lack of Specific Species Identification Is a Limitation with Currently Available Kits

Malaria species identification is important in areas where co-endemicity of non-falciparum species occurs, e.g., in areas where *P. vivax* infections—requiring additional liver-stage treatment—are prevalent [1,37]. Although the Loopamp Pf and Pv detection kits provide specific identification of *P. falciparum* and *P. vivax*, it is at an additional cost of 5.20 euros per additional test and it still does not provide a very detailed and/or accurate description of the *Plasmodium* species composition [38]. This has also shown to be of importance in areas where *P. knowlesi*—which has the ability to cause severe disease even at low parasite densities—is present. Molecular testing for quality assurance of microscopy-confirmed cases in Indonesia recently found that microscopy was unable to identify or miss-classified up to 56% of confirmed malaria cases, half of which were later determined to be *P. knowlesi* infections [66]. Although the Loopamp Pan Detection kit successfully detected these infections in dried blood spots (unlike a more standard 18S rRNA nested PCR reference targeting the four human-only species), further species identification was limited by the unavailability of species-specific testing with the platform used. This highlights the difficulties of malaria species identification at the point-of-care and reference laboratory levels in settings where co-endemicity of non-falciparum species occurs [66].

### 4.2. The Loopamp Malaria Detection Kits Provide Several Advantages in Low Resource Settings

The lower cost of the Loopamp Malaria Detection Kits, together with the endorsement from FIND, is likely the main reason as to why primarily these kits have been used in malaria endemic settings [67]. The simple and cheap boil and spin extraction method that can be used together with the Loopamp kits reduces the costs of extraction and can be performed near point-of-care reducing the time to result. A downside with the boil and spin method is that the extracted DNA is most likely not suitable for conventional PCR due to the presence of *Taq* polymerase inhibitors such as hemoglobin [4], and is not recommended for freezing due to the instability of the DNA [36,42,68]. However, the Loopamp assay has successfully been conducted on finger prick blood samples collected in extraction buffer and stored at −80 °C for up to one year before the boil and spin extraction was performed [49].

An additional advantage with the Loopamp kits is the possibility of using dried blood spots on filter paper, as supposed to venous blood samples or whole blood from finger pricks [27,28]. This allows easy transport and storage of samples at room temperature, allowing the assays to be conducted at a central laboratory rather than at point-of-care. Furthermore, if biological material is stored on filter papers, then LAMP positive samples can easily be transferred to more sophisticated laboratory facilities for more detailed analysis by PCR, e.g., for *Plasmodium* species identification and parasite density quantification.

### 4.3. Improving Throughput Will Aid Large Prevalence Surveys

Improved throughput for screening of larger numbers of samples simultaneously would benefit routine prevalence surveys in areas aiming at malaria elimination [38,42]. High-throughput performance primarily depends on the DNA extraction capacity, which is considered the main bottleneck with large number of samples [47]. The centrifugation-free methods available for both commercial kits have shown good clinical sensitivity, but at a greater cost per sample and they are not compatible with testing of large number of samples [47]. The high throughput extraction platform developed for use with the Loopamp kits requires highly specific equipment [43], without significantly improving the time-to-results compared to boil and spin extraction of the same number of samples [41]. Simplifying the sample processing protocol, e.g., by reducing the number of transfer steps, may increase throughput. This would also reduce the amount of plastic consumables needed, which is not only a benefit from an environmental point-of-view [19], but also given that sterile filter tips suitable for molecular assays are a commodity that are not readily available in low-resource-settings [64].

Finally, the use of the naked eye or a UV-lamp for the read-out of LAMP results is comparably less objective than the use of specialized equipment such as the Illumipro-10™ incubator or the LA-500 real-time turbidimeter [26,36,41,65]. However, these instruments have limited capacity of 10 or 16 samples per run. Providing a more objective format for result readout on a larger scale could be a useful, especially for use in low prevalence areas where the occurrence of positive samples is rare and might be missed.

## 5. LAMP for Malaria Screening in Pregnancy

### 5.1. LAMP for Point-of-Care Malaria Screening in Antenatal Care Programmes

Malaria in pregnancy affects the health of the fetus, resulting in substantial adverse neonatal morbidity and mortality. The WHO therefore recommends intermittent preventive treatment in pregnancy (IPTp) in areas of moderate to high malaria transmission in sub-Saharan Africa [69]. In lower transmission settings where IPTp is not implemented, accurate diagnosis and early treatment of malaria in pregnancy are crucial for preventing malaria-related pregnancy- and birth complications [70]. Low-density malaria infections that are missed by conventional diagnostic tools are, however, a prominent feature in pregnancy. Subpopulations of *P. falciparum* parasites typically accumulate in the placenta, whereas the parasite density in the peripheral circulation is low. Low-density infections are also common for *P. vivax*, with dormant hypnozoite stages that may cause multiple relapses since the radical cure primaquine is contradicted during pregnancy [71,72].

LAMP has shown to greatly improve the detection of these low-density infections in maternal peripheral blood during pregnancy [70,73,74,75,76,77] as well as in placental blood at delivery [77,78,79], when compared to microscopy, and conventional or highly sensitive RDTs (Table 4). The fact that antenatal care is usually provided in clinics where the basic laboratory procedures can easily be conducted, together with the improved sensitivity for the diagnosis of gestational and placental malaria [70,74,75,77], suggests that LAMP could provide a valuable tool in the screening of malaria during antenatal visits or at delivery. However, the relevance of detecting and treating sub-patent malaria infections in pregnant women, and the impact this may have on birth outcomes, needs to be further evaluated in larger studies [54,75]. In addition, the commercially available LAMP kits are currently retailing at a higher cost-per-test than RDT and microscopy, potentially limiting their use in low and middle-income malaria endemic countries [75,77]. Even with an affordable LAMP test, switching strategies to screening with LAMP may face additional operational and financial difficulties [75]. Consequently, further studies are needed to evaluate the cost-effectiveness of the potential integration of LAMP into maternal health programs in different transmission settings [70,77].

### 5.2. Malaria Detection by LAMP as a Surrogate for Adverse Birth Outcomes in Clinical Trials

Given their improved sensitivity, commercially available LAMP kits have recently been introduced as an additional parameter in clinical trials for assessing the effectiveness of malaria preventative measures during pregnancy [71,78,79]. The use of malaria-specific outcomes as surrogate measures of adverse birth outcomes is a common practice in such clinical trials, as they can greatly reduce sample size requirements [81]. However, there is no consensus for which malaria-specific outcomes should be used, and there is a need to develop and evaluate standardized approaches in different epidemiological settings [82].

It is currently understood that placental malaria is a major cause of adverse birth outcomes [73,80,82]; however, there are conflicting findings on the optimal method for detecting placental malaria at delivery [73,81]. One study showed that the presence of parasites in the placenta as a measure of an active infection, diagnosed by either placental blood smear, LAMP of placental blood, or parasites observed on histopathology, was significantly associated with pre-term birth, with increased risk for low birth weight and small for gestational age [73]. LAMP was the most sensitive measure of detecting malaria parasites in the placenta at delivery, suggesting that LAMP could provide the optimal diagnostic test for placental malaria. A second study, on the other hand, showed the direct opposite, that the detection of malaria parasites in the placenta by microscopy or LAMP was not associated with adverse birth outcomes, whilst presence of malaria pigment detected by histopathology—as a measure of past infection—was [81]. The placenta or placental blood, however, only becomes available at the time of delivery.

Following this, some studies have assessed the relationships between longitudinal measures of submicroscopic parasitemia during pregnancy and placental malaria. Although data are limited, current findings have shown that women with only submicroscopic infection have an increased risk of placental malaria compared to women without any parasitemia [79]. The risk for placental malaria is greater in primigravid women and increases with both frequency of infection as well as parasite density during pregnancy [79,80,82]. These findings support the use of highly sensitive diagnostic tools for identifying and targeting these infections early on during pregnancy, especially in primigravid women.

## 6. Malaria LAMP in Returning Travelers in Non-Endemic Settings

The current recommendations for malaria testing in returning febrile travelers issued by theUnited States Center of Disease Control, is to provide a preliminary result by RDT followed by on average three consecutive thick and thin films spaced 6 to 8 h apart, to ensure that no parasites are present [34]. Repeated testing of patients without malaria—based on existing algorithms in non-endemic settings where the prevalence of malaria in returning travelers is relatively low—entails a significant burden on labor in both the laboratory and repeated emergency room use, resulting in additional laboratory and hospital costs [83]. Furthermore, malaria diagnosis in non-endemic settings can be challenging as physicians and laboratory personnel outside the central care centers may have limited experience and access to gold standard testing [67]. Skillful malaria microscopy is difficult to maintain in routine diagnostic laboratories without a focus on tropical diseases, and laboratory technicians may not be experienced with reading the thick films required to ultimately rule out infection and instead rely on RDTs. The high negative predictive value of LAMP on the other hand, has shown in several studies to be able to rule out malaria, with a faster turnaround time and without the need for repeated testing [84].

The Illumigene Malaria LAMP test has primarily been evaluated for use in non-endemic areas, both prospectively on blood collected at the primary visit [30,32,83,84,85,86] or retrospectively where also follow up samples after treatment were included [30,34,83,87] (Table 5). The Loopamp Malaria Pan and Pf Detection Kits have also been evaluated in a few studies for prospective [40,88] and retrospective [67] detection among travelers in non-endemic areas. Several prospective studies comparing LAMP with microscopy and RDT have shown that most febrile patients with *P. falciparum* malaria have high enough parasite densities to be detected by microscopy and RDT when attending health care in non-endemic areas [30,83,84,85,88]. However, non-immune travelers may develop clinical disease at very low parasite densities [30,89] and one study conducted at the Institute of Tropical Medicine, Belgium showed that approximately 10% of patients with *P. falciparum* infections presented with parasite density below 100 p/µL, i.e., in the range of the detection limit of microscopy and RDT [90]. Furthermore, RDTs have a markedly higher detection limit for non-falciparum malaria, and these species—especially *P. ovale* and *P. malariae*—typically present with low parasite densities and are therefore often missed [90].

The practicality of the Illumigene kit and accompanying equipment has proven to be useful in non-endemic settings [32]. It has a high negative predictive value for all species when comparing against PCR [86], and has therefore been suggested for use in screening of imported malaria in non-endemic countries when expert microscopists are not immediately available. However, the rare occurrence of in-valid results [32,34], and the need for species identification and parasite quantification, precludes the use of the Illumigene Malaria test as single reference method in the diagnosis of imported malaria [32]. Nevertheless, the Illumigene platform has greater analytical sensitivity than microscopy, even in specialized settings. Hence, the introduction of LAMP into existing algorithms in non-endemic settings—as an alternative to RDTs in the point-of-care screening prior to blood film microscopy—could be recommendable [84].

A cost analysis assessment of a novel algorithm for screening febrile patients with LAMP in a non-endemic setting, estimated that a single LAMP test with a high negative predictive value will provide a per-patient cost saving of USD$13 [83]. Despite an increase in material costs with the LAMP algorithm, the labor and hospital costs incurred by repeat microscopy on negative initial results are dramatically reduced [34]. The cost-effectiveness of introducing LAMP will, however, depend on the number of samples, the positivity rate, and the experience of the microscopist, and must be validated in every laboratory independently [30].

## 7. Expert Opinion

The need for highly sensitive, field adaptable diagnostic tools that can readily be used in low resource settings is apparent [51]. Although LAMP has been suggested as a promising tool for use in malaria control and elimination programs [25], its use has primarily been limited to research settings to date [24]. The question remains if and how LAMP will optimally be employed in the future of malaria diagnosis, either as a near point-of-care diagnostic tool or as a high-throughput surveillance tool to support malaria elimination activities [9]. The answer will very much depend on available resources, malaria transmission levels, laboratory equipment needs, and staff training requirements [24].

Although LAMP in its current form and cost is unlikely to become a reality for point-of-care diagnosis of clinically symptomatic malaria in low resource settings [1,9,33,44], it could improve the detection and management of malaria early on in pregnancy by incorporating LAMP screening into antenatal care programs where IPTp is not implemented [70,74,75,77]. However, the impact that detecting and treating sub-patent *P. falciparum* infections in pregnant women might have on adverse birth outcomes requires further evaluation [73,80,81], together with thorough assessment of the cost effectiveness of the integration of LAMP as a tool for malaria screening during pregnancy in different transmission settings [70,77]. Furthermore, given its high negative predictive value, malaria LAMP can rule out malaria infection making it a cost-effective alternative for screening of returning febrile travelers [30,34,83]. However, malaria LAMP in its current state will not be able to replace the use of microscopy in non-endemic settings, which will still be required for parasite density quantification and further species identification in malaria positive specimens [91].

Malaria LAMP may also provide a powerful molecular alternative to enable the detection of low-density infections especially in areas where prevalence is declining [9]. While there is some discussion about the clinical relevance of detecting low density and asymptomatic infections, it is clear that they play an important role in malaria transmission in low transmission areas [1,51]. Although the analytical sensitivity of LAMP is in line with the WHO recommendation that molecular-based diagnostic tools for use in low transmissions settings should achieve an analytical sensitivity of at least two parasites per microliter [54], the question remains if this will be sufficient [51]. Studies conducted in asymptomatic and low-density infections have shown a reduced sensitivity when parasite densities go below two parasites per microliter. This is likely to be of significance in the evaluation of malaria active and reactive elimination strategies in low transmission settings, where a significant proportion of asymptomatic infections may fall beneath this limit [41,46,50,52,92,93]. Furthermore, thorough assessments of the cost per assay, including equipment, reagents, labor, training and maintenance are needed to evaluate the cost-effectiveness of malaria LAMP in different transmission settings [1,9,70,94].

### Future Research Directions

Following is a suggested list of areas that require further exploration and/or development for commercially available LAMP kits:Improving *Plasmodium* species differentiation, especially for *P. knowlesi.*Improving the throughput of commercial kits, whilst providing objective read-out of results, and reducing the need for plastic consumables.Assessing the impact on transmission of applying LAMP in malaria elimination strategies.Assessing the impact on adverse birth outcomes of early detection of low-density malaria infections by LAMP during pregnancy.Detailed assessments of cost and cost-effectiveness of malaria LAMP in different transmission settings, especially for use in malaria elimination and antenatal care programs.

## 8. Conclusions

Despite LAMP being a promising near point-of-care tool for use in low resource settings, it is unlikely that commercially available LAMP kits will replace conventional diagnostic tools in the diagnosis of clinical malaria in endemic areas. They may, however, provide a sensitive molecular alternative for use in malaria elimination programs, as well as in malaria screening during pregnancy in low transmission settings, and in returning travelers in non-endemic settings. Nevertheless, further improvements of the already available commercial kits and detailed assessments of cost-effectiveness could improve their uptake on a larger scale.

## Figures and Tables

**Table 1 diagnostics-11-00336-t001:** Pooled estimates of the diagnostic accuracy of malaria loop-mediated isothermal amplification as determined in three meta-analyses.

Comparator	Sensitivity (%; 95% CI)	Specificity (%; 95% CI)	AUC	Reference
LAMP vs. LM	98; 94–99	97; 85–99	ND	[1]
LAMP vs. PCR	96; 79–99	91; 68–98	ND	[1]
LAMP vs. LM	97; 96–98	96; 94–97	0.98	[24]
LAMP vs. RDT	97; 92–99	96; 92–98	0.98	[24]
LAMP vs. PCR	97; 96–98	96; 94–97	0.98	[24]
*Pv* LAMP vs. PCR	95; 80–99	96; 86–99	0.98	[24]
Pan LAMP vs. PCR	95; 91–97	98; 95–99	0.99	[25]
*Pf* LAMP vs. PCR	96; 94–98	99, 96–100	0.99	[25]
*Pv* LAMP vs. PCR	98; 92–99	99, 72–100	1.00	[25]

95% CI: 95% confidence interval; AUC: area under the curve; LAMP = loop-mediated isothermal amplification; LM = light microscopy; PCR = polymerase chain reaction; RDT = rapid diagnostic test; *Pv* = *P. vivax*; Pan = Pan-*Plasmodium*; *Pf* = *P. falciparum*; ND = not determined.

**Table 2 diagnostics-11-00336-t002:** Comparison between the Illumigene^®^ (Alethia^®^) malaria LAMP (Meridian Bioscience Inc.) and the Loopamp^TM^ Malaria Pan Detection kits (Eiken Chemical Co.).

Comparator	Illumigene^®^	Loopamp^TM^
Species identification	Pan-*Plasmodium*	Pan-*Plasmodium*; *P. falciparum*; *P. vivax* ^1^
Sample type	Fresh/frozen blood	Fresh/frozen blood or dried blood spots
Methods of DNA extraction	Illumigene malaria; Illumigene malaria PLUS	PURE DNA Extraction Kit; Boil and spin; other methods ^2^
Limit of detection ^3^	2.0 p/µL for *P. falciparum* and 0.1 p/µL for *P. vivax*	1–2 p/µL
Required equipment	Illumipro-10™ incubator	Centrifuge, heat block/water bath, UV light, (or LA-500 turbidimeter/HumaLoop M)
Number of samples per run	10	16, 46 or 94 ^4^
Read out of results	Turbidity in Illumipro-10™ incubator or by eye	Turbidity in turbidimeter or by eye; fluorescence under UV light
Primary area of use	Malaria diagnosis in non-endemic settings	Malaria prevalence surveys in endemic settings
Cost per test ^5^	28 EUR	5.2 EUR

^1^ Three separate kits. ^2^ Including Chelex extraction, column-based extraction methods, and a high-throughput DNA extraction platform. ^3^ Under laboratory conditions. ^4^ If used together with a turbidimeter or HumaLoop M platform, regular heating block, or high-through put extraction setup, respectively. ^5^ Does not include cost for equipment.

**Table 3 diagnostics-11-00336-t003:** Performance of malaria loop-mediated isothermal amplification for detecting asymptomatic low-density infections.

Setting	Sample Size ^1^	Sample Type ^2^	LAMP Method	Comparator	Prevalence (%)	Mean Parasite Densities (p/µL)	Sensitivity (%; 95% CI)	Specificity (%; 95% CI)	Reference
Zanzibar, Pre-elimination	996	Fresh blood + Boil and Spin	Loopamp Pan/Pf	RDT; Ref: qPCR	RDT: 1.0 LAMP/PCR: 1.8	26 (range: 0–4626).	Pan LAMP: 83.3; 59–96 RDT: 55.6; 31–79	Pan-LAMP: 99.7; 99–100 RDT: 100; 99.6–100	[38]
Zanzibar, Pre-elimination	3983	Fresh blood + Boil and Spin	Loopamp Pan/Pf	RDT; Ref: LAMP	RDT: 0.5; LAMP 1.6	ND; 71% of LAMP positives <LOD of RDT	RDT: 24.6; 15–37	RDT: 99.9; 99.7–100	[42]
Zanzibar, Pre-elimination	3008	Filter device +HTP extraction	Loopamp Pan/Pf	RDT; Ref: qPCR	RDT: 0.4; qPCR: 1.6; HTP-LAMP 0.7	1.8 (range: 0.1–770)	HTP-LAMP: 40.8; 27–56 Chelex-LAMP: 49; 34–64 HTP_LAMP >2: 54; 25–81 HTP_LAMP ≤2: 36; 21–54	HTP-LAMP: 99.9; 99.8–100	[41]
Eswatini (formally Swaziland), Very low transmission	10890	DBS + Chelex extraction	Loopamp Pan/Pf	RDT Ref; LAMP LAMP Ref: nPCR	RDT: 0.6; LAMP: 1.7	ND; 67% of LAMP positives <LOD of RDT	LAMP: 72.2; 63−80 RDT: 33.4; 33–35	LAMP: 98.0; 97−98 RDT: >90.0	[52]
Namibia, Low transmission	2642	DBS and used RDTs + Chelex extraction	Loopamp Pan	RDT Ref: nPCR	RDT: 0.9; LAMP 1.8	ND; 51% of LAMP positives <LOD of RDT	LAMP on RDT: 95.4; 84–99 LAMP on DBS: 95.5; 85–99 RDT: 9.3; 2.6–22	All > 99	[48]
Colombia, Varied transmission	980	Fresh blood + Boil and Spin	Loopamp Pan/Pf	LM Ref: qPCR	LM: 0.2; LAMP: 6.6; qPCR: 7.2	ND, (range: 1–897)	Pv: 90.9; 80–97 Pf: 100; 78–100	All > 99	[53]
Peruvian Amazon, Low to moderate transmission	1167	Fresh blood + Boil and Spin	Loopamp Pan/Pf	LM Ref: qPCR	LM: 4.9; LAMP: 21.9	10 (CI95% 7.5–13)	LAMP: 91.8; 88–95 LM: 20.3; 16–26	91.9; 88–95 LM: 98.0; 95–99	[47]
Uganda, High transmission	554	DBS + Chelex extraction	Loopamp Pan	LM Ref: qPCR	LM: 18.2; LAMP: 37.2; qPCR: 48.9	LAMP neg: 0.1 (CI95% 0.07–0.2) LAMP pos: 5.7 (CI95% 3.0–10.8)	LAMP: All LM negs: 44.7 ≥0.01–<0.1 p/µL: 10.8 ≥0.1–<1 p/µL: 40.9≥1 p/µL: 81.5	All LM negs: 94.0	[46]

^1^ All asymptomatic; ^2^ All samples from finger prick; p/µL = parasite per microliter; 95% CI = 95% confidence interval; Ref = reference method; Pf = *P. falciparum*; Pan = Pan-*Plasmodium*; RDT = rapid diagnostic test; LM = light microscopy; qPCR = quantitative PCR; nPCR = nested PCR; LAMP = loop-mediated isothermal amplification; DBS = dried blood spot; HTP = high through put; <LOD = below the limit of detection; ND = not determined.

**Table 4 diagnostics-11-00336-t004:** Performance of loop-mediated isothermal amplification for detecting malaria during pregnancy or at delivery.

Setting	Sample Size	Sample Type	LAMP Method	Comparator	Prevalence (%)	Sensitivity (%; 95% CI)	Specificity (%; 95% CI)	Reference
Colombia	531	Venous and placental blood	Loopamp Pan/Pf	LM + RDT; Ref: nPCR	Peripheral: LM 5.8; RDT 5.6; LAMP 7.3; nPCR 7.3 Placental: LM + RDT 0.8; LAMP 3.1; nPCR 3.6	Peripheral: LAMP: 100; 92–100 LM: 80; 65–89 RDT: 77; 62–87 Placental: LAMP: 89; 57–98 LM+RDT: 22; 6–55	100% all tests	[70]
Colombia	858	Finger prick blood	Loopamp Pan/Pf	LM; RDT; hsRDT; nPCR; Ref: qRT-PCR	LM 2.7; RDT 2.4; hsRDT 3.0; LAMP 5.2; nPCR 4.2 qRT-PCR 5.5	LAMP: 90; 76–97 LM: 59; 42–74 RDT: 54; 37–70 hsRDT: 64; 47–79 nPCR: 77; 61–89	≥99.9% all tests	[77]
Ethiopia	87	Venous blood	Loopamp Pan/Pf	LM + RDT; Ref: nPCR	LM 11.5; RDT 10.3; nPCR 11.5 LAMP 17.2	LAMP: 100 LM: 90; 66–100 RDT: 70; 34–100	LAMP: 94; 87–100 LM: 99; 97–100 RDT: 97; 93–100	[74]
Ethiopia	193	Venous blood	Illumigene	LM + RDT; Ref: LAMP	LM 2.0, RDT 2.0; LAMP 4.2	LM: 56; 21–86 RDT: 67; 30–93	100 both tests	[75]
Uganda	282	DBS from peripheral, placental and cord blood	Loopamp Pf	LM + HP at delivery	Placental: LM 2.9; LAMP 8.6; Peripheral: LM 2.1; LAMP 10.0; (Cord: LM 0.0; LAMP 1.1; HP: 37.2)	ND	ND	[73,78,80]
Uganda	687	DBS from placental blood	Loopamp Pf	LM + HP at delivery	Placental: LM 4.4; LAMP 12.0; HP: 44.6	ND	ND	[79,81]

95% CI = 95% confidence interval; Pf = *P. falciparum*; Pan = Pan- *Plasmodium*; DBS = dried blood spot; ANC = antenatal care; Ref = reference method; LAMP = loop-mediated isothermal amplification; LM = light microscopy; RDT = rapid diagnostic test; nPCR = nested PCR; hsRDT = highly sensitive RDT (Pf only); qRT-PCR = quantitative reverse transcriptase PCR; HP = histopathology (parasites or pigment); ND = not determined.

**Table 5 diagnostics-11-00336-t005:** Performance of malaria loop-mediated isothermal amplification in returning travelers in non-endemic settings.

Setting	Sample Size	Sample Type ^1^	LAMP Method	Comparator	Sensitivity (%; 95% CI)	Specificity (%; 95% CI)	NPV (%)	PPV (%)	Reference
Belgium	133	Pro + Retrospective	Illumigene	qPCR	100; 95–100	100; 90–100	ND	ND	[30]
Canada	140	Retrospective	Illumigene	LM + qPCR	97.3; 91–100	93.8; 85–98	99.8	45.2	[34]
France	310	Prospective	Illumigene	qPCR	100; 96–100	98.1; 95–99	100	95.5	[32]
Canada	348	Pro + Retrospective	Illumigene	LM + qPCR	100; 94–100	100; 99–100	100	100	[83]
Germany	1000	Prospective	Illumigene	LM Ref: qPCR	98.7 LM: 76.1	99.6 LM: 100	99.6 LM: 86	98.7 LM: 100	[84]
Italy	478	Prospective	Illumigene	LM + RDT Ref: qPCR	100 LM: 94.7 RDT: 92.1	100 LM: 100 RDT: 100	100 LM; 99.0 RDT: 98.5	100 LM: 94.7 RDT: 100	[85]
Denmark	38	Retrospective	Illumigene	qPCR	96.4	ND	ND	ND	[87]
France	331	Prospective	Illumigene	LM + RDT Ref: qPCR	97.3 LM: 84.9 RDT: 86.3	99.6 LM: 99.6 RDT: 100	99.8 LM: 98.9 RDT: 99.0	94.8 LM: 94.1 RDT: 100	[86]
Great Britain	705	Prospective	Loopamp Pan/Pf	LM; Ref: nPCR	97.0; 90–100 LM: 84; 73.92	99.2; 98–100 LM: 100; 99–100	99.7 LM: 98.3	92.7 LM 100	[40]
Switzerland	205	Prospective	Loopamp Pan/Pf	qPCR	100; 92–100	100; 98–100	100	100	[88]
Canada	140	Retrospective	Loopamp Pan/Pf	RDT; Ref: nPCR	100; 93–100 RDT: 85.9; 75–92	98.6; 91–99 RDT: 98.6; 91–100	ND	ND	[67]

^1^ All venous blood samples. 95% CI = 95% confidence interval; LAMP = loop-mediated isothermal amplification; NPV = negative predictive value; PPV = positive predictive value; LM = light microscopy; qPCR = quantitative PCR; nPCR = nested PCR; Ref = reference method; ND = not determined.

## Data Availability

No new data were created or analyzed in this study. Data sharing is not applicable to this article.

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
