# Peer review of "Performance and Application of Commercially Available Loop-Mediated Isothermal Amplification (LAMP) Kits in Malaria Endemic and Non-Endemic Settings"

_diagnostics, 2021, doi:10.3390/diagnostics11020336_

Round 1

Reviewer 1 Report

The manuscript presented by Morris and Aydin-Schmidt presents a review of the literature regarding the use of LAMP in the laboratory diagnosis of malaria. The manuscript is very well prepared, since it presents the main advantages and disadvantages of technology (LAMP) as well as the contextualization of the different aspects of the problem of malaria, especially the different epidemiological scenarios of the disease.

A minor suggestion / comment:
- A schematic representation of how LAMP technology works would be important for readers in other areas.

Author Response

A minor suggestion / comment:
- A schematic representation of how LAMP technology works would be important for readers in other areas.

Reply: We thank the reviewer kindly for this suggestion. The molecular process of LAMP amplification has been schematically presented else where (see reference 2 in the Manuscript (Notomi, et al.. 2000)), hence we have chosen not to add a schematic representation to this review.

Reviewer 2 Report

The authors reviewed the performance of two commercially available LAMP kits for malaria diagnosis used in different applications: detection of low parasitemia, diagnosis in pregnant women, and diagnosis of imported malaria. LAMP technology is a highly promising diagnostic tool for many infectious diseases, including malaria, in resource-poor countries. It is more sensitive than microscopy or rapid diagnostic test for malaria.

The background section provides relevant information. The advantages and disadvantages of each LAMP kit are well described in detail. The tables (tables 3–5) provide a concise and useful presentation of LAMP performance in different contexts.

The paper is highly pertinent and timely for further development and application of this innovative technology, especially in the field.

The text is clearly written. The review is organized according to the format of the series of Taylor & Francis journals “Expert Opinion on …” The last section (Expert opinion) is the authors’ reflections on the future direction of where LAMP should be heading to and is helpful for readers.

Below are few minor suggestions to improve the paper. Please provide the full name the first time when abbreviations are used. Please also use the same format for the references.

Major comments:

none

Minor comments:

Lines 37-38: “The design of the primers resultS in…”

Line 52: some of which were recently reviewed

Line 78, Table 1: 95% CI (instead of CI95%); Please add what “95% CI” stands for in the legend (lines 79-81). Pan-Plasmodium (capital letter “P”)

Line 85: Some international readers may not know what CE stands for. Please define it (abbreviation of “Conformité Européenne” in French or “European Conformity”).

Line 112: the detection limit of the assay is…

Lines 122-123: …the Foundation who has developed

Line 143: …platform…has been evaluated/assessed (instead of trialled)

Line 144: …kits cost about… 5.20 euros per test

Table 2: 3rd column, P. vivax

Line 156, “some 30 publications”: Please cite the exact number of publications. Suggest: The use of LAMP has been evaluated in 30 publications

Line 161: minimum of laboratory training

Line 187: all of which were conducted in sub-Saharan Africa

Line 212: There seems to be something missing in this sentence, from “if clinical malaria…”

Lines 220-221: by technicians with enough skills; comma (,) instead of a semi-colon (;) in line 221

Line 233: severe disease

Line 281: antenatal

Lines 285-286: diagnosis and treatment are crucial

Lines 291: radical cure with primaquine is contraindicated; Reference 73: A more relevant reference on the contraindication of primaquine during pregnancy should be cited here (for example, WHO Guidelines for the treatment of malaria. Third edition, 2015).

Line 306: surrogate (spelling); birth (space) outcomes; clinical (spelling) trials

Line 343: skillful

Lines 368-371: Please check the syntax. (delete “and” then “introduction of the use of LAMP…)

Table 3 legend: 95%CI = 95% confidence interval; Pf = P. falciparum (in italics)

Line 433: “commercially available LAMP kits to become more readily available” It may be better to replace the second “available” with an alternative adjective because LAMP kits are already ‘readily available’ for those who have sufficient budget.

Lines 448-449: “Although further improvements…on a larger scale” This is a subordinate clause without an independent clause attached to it. Should the period after “non-endemic settings” (line 447) be a comma? Please write a complete sentence.

Reference 3, 11: Please complete these references (volume, page numbers).

Reference 13: Please delete “the official publication…Infectious Diseases”

References 14, 17, 21, 26, 31, 34, 41, 44, 46, 49, 50, 52, 57, 58, 61, 64, 65, 79, 81, 82, 84, 87: They are not in the same format as the other references (see article titles, first letters of each word in capital letter).

Reference 24: Please delete “: IJID: official publication…Infectious Diseases”

Reference 26: volume, page numbers

Reference 50: Trends Parasitol 2020;36(11):898-905.

References 52, 88, 91: Please delete “: an official publication…Infectious Diseases Society of America”

References 55, 56, 71: Please add the links.

Reference 86: Please add the page numbers or article number (taz052) after the volume number.

Author Response

Please provide the full name the first time when abbreviations are used. Please also use the same format for the references.

Reply: We have checked though the manuscript for abbreviations and made corrections for example on lines 17, 34, 38, 44, 51, 89, 219, 341 in the revised manuscript. We have also corrected the format of the references, however, some of these formatting was done in Endnote, hence it was not possible to track all the changes.

Minor comments:

Lines 37-38: “The design of the primers resultS in…” Reply: Corrected as per suggestion.

Line 52: some of which were recently reviewed Reply: Corrected as per suggestion.

Line 78, Table 1: 95% CI (instead of CI95%); Please add what “95% CI” stands for in the legend (lines 79-81). Pan-Plasmodium (capital letter “P”) Reply: Corrected as per suggestion.

Line 85: Some international readers may not know what CE stands for. Please define it (abbreviation of “Conformité Européenne” in French or “European Conformity”). Reply: Corrected as follows on line 89 of the revised manuscript: “CE-marked (where CE stands for Conformité Européenne” in French or “European Conformity” in English) malaria LAMP kits”.

Line 112: the detection limit of the assay is… Reply: Corrected as follows on line 117 of the revised manuscript: “The detection limit of the assay is 2.0 parasites per microliter….”.

Lines 122-123: …the Foundation who has developed. Reply: Corrected as per suggestion.

Line 143: …platform…has been evaluated/assessed (instead of trialled). Reply: Corrected to “assessed” as per suggestion.

Line 144: …kits cost about… 5.20 euros per test. Reply: Corrected to “The Loopamp kits cost about 5.20 euros per test” as per suggestion.

Table 2: 3rd column, P. vivax. Reply: Corrected as per suggestion.

Line 156, “some 30 publications”: Please cite the exact number of publications. Suggest: The use of LAMP has been evaluated in 30 publications Reply: Corrected as follows on line 161 of the revised manuscript: “The performance of LAMP has been evaluated in over 30 publications in malaria endemic field settings, ..”

Line 161: minimum of laboratory training Reply: Corrected as follows on line 167 of the revised manuscript: “…the methods are easy to perform after only 3-5 days laboratory training ….”.

Line 187: all of which were conducted in sub-Saharan Africa Reply: Corrected as per suggestion.

Line 212: There seems to be something missing in this sentence, from “if clinical malaria…” Reply: Corrected as follows on line 218 of the revised manuscript: “It is unlikely that LAMP will replace conventional diagnostic tools such as RDTs and microscopy in the point-of-care diagnosis of clinically symptomatic malaria in endemic settings”.

Lines 220-221: by technicians with enough skills; comma (,) instead of a semi-colon (;) in line 221 Reply: Corrected as follows on line 225-227 of the revised manuscript: “Despite the ease of use, LAMP still requires good enough facilities for preparing basic molecular assays, ideally with separate workstations for avoiding DNA-cross-contamination [64]. In addition, the technicians running the assays require sufficient knowledge to manage DNA contamination, an issue with all nucleic acid amplification-based methods that needs to be addressed”.

Line 233: severe disease Reply: Corrected as per suggestion.

Line 281: antenatal Reply: Corrected as per suggestion.

Lines 285-286: diagnosis and treatment are crucial Reply: Corrected as per suggestion.

Lines 291: radical cure with primaquine is contraindicated; Reference 73: A more relevant reference on the contraindication of primaquine during pregnancy should be cited here (for example, WHO Guidelines for the treatment of malaria. Third edition, 2015). Reply: The suggested refence [Ref nr 73] has been added to line 299 in the revised manuscript.

Line 306: surrogate (spelling); birth (space) outcomes; clinical (spelling) trials Reply: Corrected as per suggestion.

Line 343: skillful Reply: We have not corrected this as we have used British English throughout the manuscript.

Lines 368-371: Please check the syntax. (delete “and” then “introduction of the use of LAMP…) Reply: Corrected as follows on line 374-377 of the revised manuscript: “Nevertheless, the Illumigene platform has greater analytical sensitivity than microscopy, even in specialized settings. Hence, the introduction of LAMP into existing algorithms in non-endemic settings —as an alternative to RDTs in the point-of-care screening prior to blood film microscopy— could be recommendable”.

Table 3 legend: 95%CI = 95% confidence interval; Pf = P. falciparum (in italics) Reply: Corrected as per suggestion.

Line 433: “commercially available LAMP kits to become more readily available” It may be better to replace the second “available” with an alternative adjective because LAMP kits are already ‘readily available’ for those who have sufficient budget. Reply: Corrected as follows on line 439-440 of the revised manuscript: “Following is a suggested list of areas that require further exploration and/or development for commercially available LAMP kits:”.

Lines 448-449: “Although further improvements…on a larger scale” This is a subordinate clause without an independent clause attached to it. Should the period after “non-endemic settings” (line 447) be a comma? Please write a complete sentence. Reply: Corrected as follows on line 455-456 of the revised manuscript: “Nevertheless, further improvements of the already available commercial kits and detailed assessments of cost-effectiveness could improve their uptake on a larger scale.”.

Reference 3, 11: Please complete these references (volume, page numbers). Reply: Ref 3 corrected in the revised manuscript, also removed ref 54 in original manuscript as it was a duplication of reference 3. Ref 11 is online ahead of print, I could not find the format for this in the author guidelines.

Reference 13: Please delete “the official publication…Infectious Diseases” Reply: Corrected Ref 15 in the revised MS.

References 14, 17, 21, 26, 31, 34, 41, 44, 46, 49, 50, 52, 57, 58, 61, 64, 65, 79, 81, 82, 84, 87: They are not in the same format as the other references (see article titles, first letters of each word in capital letter). Reply: We updated the Endnote style file recommended by the ACS style guide in the author guidelines for the title capitalization to be “Sentence style captilization”, all these reference should therefore be corrected now but without track changes.

Reference 24: Please delete “: IJID: official publication…Infectious Diseases” Reply: Corrected Ref 24 in the revised MS.

Reference 26: volume, page numbers. Reply: Corrected Ref 26 in the revised MS.

Reference 50: Trends Parasitol 2020;36(11):898-905. ” Reply: Corrected Ref 50 in the revised MS.

References 52, 88, 91: Please delete “: an official publication…Infectious Diseases Society of America” Reply: Corrected Ref 52, 88, 91 in the revised MS.

References 55, 56, 71: Please add the links. Reply: Added links to references 54, 55 and 70 in the revised manuscript.

Reference 86: Please add the page numbers or article number (taz052) after the volume number. Reply: Corrected Ref 86 in the revised MS.